# Biophysical analysis of *Plasmodium falciparum* Hsp70-Hsp90 organising protein (PfHop) reveals a monomer that is characterised by folded segments connected by flexible linkers

Stanley Makumire[1], Tawanda Zininga[1,2], Juha Vahokoski[3], Inari Kursula[3,4], Addmore Shonhai[1] *

1 Department of Biochemistry, School of Mathematical & Natural Sciences, University of Venda, Thohoyandou, South Africa, 2 Department of Biochemistry, Stellenbosch University, Stellenbosch, South Africa, 3 Department of Biomedicine, University of Bergen, Bergen, Norway, 4 Biocenter Oulu & Faculty of Biochemistry and Molecular Medicine, University of Oulu, Oulu, Finland

* addmore.shonhai@univen.ac.za

**Data Availability Statement:** All relevant data are within the manuscript and its Supporting Information files.

## Abstract

*Plasmodium falciparum* causes the most lethal form of malaria. The cooperation of heat shock protein (Hsp) 70 and 90 is thought to facilitate folding of select group of cellular proteins that are crucial for cyto-protection and development of the parasites. Hsp70 and Hsp90 are brought into a functional complex that allows substrate exchange by stress inducible protein 1 (STI1), also known as Hsp70-Hsp90 organising protein (Hop). *P. falciparum* Hop (PfHop) co-localises and occurs in complex with the parasite cytosolic chaperones, PfHsp70-1 and PfHsp90. Here, we characterised the structure of recombinant PfHop using synchrotron radiation circular dichroism (SRCD) and small-angle X-ray scattering. Structurally, PfHop is a monomeric, elongated but folded protein, in agreement with its predicted TPR domain structure. Using SRCD, we established that PfHop is unstable at temperatures higher than 40°C. This suggests that PfHop is less stable at elevated temperatures compared to its functional partner, PfHsp70-1, that is reportedly stable at temperatures as high as 80°C. These findings contribute towards our understanding of the role of the Hop-mediated functional partnership between Hsp70 and Hsp90.

## 1. Introduction

Heat shock proteins (Hsp) serve primarily as protein folding facilitators. They also participate in several other processes, such as protein transport, assembly/disassembly of protein complexes, protein degradation, amongst others [1]. Their role in the survival and pathogenicity of malaria parasites is increasingly becoming apparent [2, 3, 4]. *Plasmodium falciparum* is the agent for the most lethal form of malaria. It has been reported that the cytosolic *P. falciparum* heat shock protein 70–1 (PfHsp70-1) is cyto-protective due to its ability to suppress protein mis-folding and aggregation under stressful conditions [5, 6]. In addition, another cytosolic

**Funding:** 1. IK, Sigrid Jusélius Foundation; https://sigridjuselius.fi/en/ 2. AS, L1/402/14–1; Deutsche Forchungsgemeinshaft; https://www.validate-network.org/event/german-african-cooperation-projects-in-infectiology-dfg 3. AS; 75464; 92598; National Research Foundation/Department of Science & Innovation of South Africa; https://www.nrf.ac.za/ 3. TZ; 111989; National Research Foundation of South Africa; www.nrf.ac.za

**Competing interests:** All authors have no competing interests.

molecular chaperone, *P. falciparum* Hsp90 (PfHsp90) is essential [7]. The cooperation of Hsp70 and Hsp90 is known to facilitate folding and function of proteins implicated in cell development, such as steroid hormone receptors and kinases [8, 9].

Stress inducible protein 1 (STI1) was first described in mouse [10], and now also known as Hsp70-Hsp90 organising protein (Hop), acts as a module that allows Hsp70 and Hsp90 to interact stably, thereby facilitating substrate transfer from Hsp70 to Hsp90. Interestingly, Hsp70 and Hsp90 resident in the cytosols of *E. coli* and yeast as well as their homologs localized to mouse ER were shown to interact directly independent of Hop mediation [11, 12, 13, 14]. Although *E. coli* lacks a Hop homolog [15] and the same protein is not essential in yeast [16], it has been reported to play an important role in the development of *Trypanosoma cruzi* and *Trypanosoma brucei* parasites [17]. This shows that the role of Hop varies across species and developmental stages. In light of the essential roles of the *P. falciparum* cytosol localised chaperones, PfHsp90 and PfHsp70-1, PfHop has been proposed as a potential antimalarial drug target [18, 19] This is partly because it occurs in complex with the two chaperones, and also exhibits some degree of sequence divergence from the human homolog, thus raising prospects for selective targeting by small molecule inhibitors [18]. Indeed, in some disease models such as cancer [20] and leishmaniasis [21, 22], the essential role of Hop has made it a promising drug target.

Hop is a conserved and stress inducible protein that possesses three tetratricopeptide repeats (TPR): TPR1, TPR2A and TPR2B [10]. Both Hsp70 and Hsp90 interact with Hop via the C-terminal EEVD motif, present in the two molecular chaperones [19, 23]. Hop interacts with Hsp70 and Hsp90 via its TPR1 and TPR2A domains, respectively [23]. While for a long time the role of the TPR2B domain of Hop has remained largely elusive, it is now thought that Hsp70 first binds to the TPR1 domain of Hop before switching to the TPR2B domain to facilitate substrate transfer to Hsp90 [20, 23].

In light of the importance of both PfHsp70-1 and PfHsp90 in the survival of the malaria parasite, there has been growing interest in identifying inhibitors targeting the function of these two molecular chaperones. Compounds that inhibit PfHsp70-1 [24, 25, 26] and PfHsp90 [9, 27] have been identified, and some of them exhibit antiplasmodial activity. Some compounds that target PfHsp90 function reverse parasite resistance to traditional antimalarial drugs, such as chloroquine (reviewed in [28]). We previously described *Plasmodium falciparum* Hop (PfHop), which co-localises and associates with both PfHsp70-1 and PfHsp90 [18, 19]. While Hop in other organisms, such as yeast and human, has been rather extensively characterised, the structure and function of PfHop remain to be elucidated.

Here, we show that PfHop is a monomeric, elongated but folded protein, which loses most of its secondary structure at temperatures above 40 ℃. We discuss the implications of our findings with respect to the role of PfHop in coordinating the Hsp70-Hsp90 pathway in *P. falciparum*.

## 2. Methods and materials

### 2.1 Materials

Reagents used in this study, unless otherwise stated, were purchased from Merck Chemicals (Darmstadt, Germany), Thermo Scientific (Illinois, USA), Zymo Research (USA), Melford (Suffolk, UK), and Sigma-Aldrich (USA). Nickel NTA resin was purchased from Thermo Scientific (USA). ECL was purchased from (ThermoFisher Scientific, USA). The expression and purification of his-tagged recombinant forms of PfHop was confirmed by Western blotting using anti-His antibodies (Thermo Scientific, USA). Furthermore, rabbit raised anti-PfHop

antibodies (Eurogentec, Belgium; [18]) were also used to confirm the presence of recombinant PfHop protein.

## 2.2 Expression and purification of recombinant PfHop

Recombinant PfHop (PF3D7_1434300) was overexpressed in *Escherichia coli* XL1 Blue cells and purified by nickel affinity chromatography as previously described [18, 19]. The Ni-affinity purified proteins were extensively dialysed in SnakeSkin dialysis tubing 10 000 MWCO (ThermoFisher Scientific, USA) against buffer A [20 mM Tris-HCl, pH 7.5, 10 mM NaCl, 5% (v/v) glycerol, 0.2 mM Tris-carboxyethyl phosphine (TCEP)]. The protein from Ni-NTA chromatography was used to perform conventional CD spectroscopy and tryptophan fluorescence assays. SRCD and SAXS analyses were performed using protein that was further purified using ion exchange and size exclusion chromatography as follows. The dialysed protein was further purified using anion exchange chromatography using a Tricorn MonoQ 4.6/100 PE column (G.E Healthcare LS, USA). PfHop was eluted by applying buffer B (20 mM Tris-HCl, pH 7.5, 10 mM NaCl, 0.2 mM TCEP) to the column using a linear (0.1–1.0 M) NaCl gradient. As the final purification step and to evaluate the oligomeric state of PfHop, size exclusion chromatography was used. Following anion exchange, fractions containing pure PfHop were pooled together and loaded onto a HiLoad 16/600 Superdex$^{TM}$ 200 pg column equilibrated with buffer C (10 mM Tris-HCl, pH 8, 300 mM NaCl containing 5% glycerol, 0.2 mM TCEP). Eluted fractions were analysed using SDS-PAGE to determine the purity and homogeneity of the PfHop protein. Authenticity of the purified protein was confirmed by sequencing using MALDI-TOF mass spectrometry at the Biocenter Oulu Proteomics Core Facility, Oulu University, Finland. The protein concentration was determined by measuring the UV absorbance at 280 nm using a Nanodrop ND100 (ThermoFisher Scientific, USA).

The molecular weight of PfHop was determined using multi-angle static light scattering (MALS) coupled to size exclusion chromatography using a Superdex S200 Increase 10/300 GL column (GE Healthcare). The column, equilibrated with buffer C, was coupled to a mini DAWN TREOS MALS detector (Wyatt Technology, Germany) and an ERC RefraMAx520 differential refractometer (ERC, Germany). 100 μg of PfHop in buffer C was injected into the column using a flow rate of 0.5 ml/min. BSA and ovalbumin were used as molecular-weight controls. The molecular weight of PfHop was determined based on the measured light scattering at three different angles and the refractive index using the ASTRA software version 6.1.5.22 (Wyatt Technology, Germany).

## 2.3 Investigation of the secondary structure of PfHop

The secondary structure of PfHop was investigated using synchrotron radiation (SR) and conventional circular dichroism (CD) spectroscopy. The spectral measurements were conducted at the UV-CD12 beam line (Anka, Karlsruhe) under temperature-controlled conditions. PfHop at a concentration of 0.5 mg/ml dialysed in buffer D (10 mM K$_3$PO$_4$, pH 7.0, 150 NaF) was analysed using a 98.56 μm path length round cell cuvette (Suprasil, Hellma Analytics, Germany) at a constant temperature of 10˚C. A total of 3 full spectral scans were recorded from 280 to 175 nm and averaged. CD spectroscopy experiments were done using a Jasco J-1500 CD spectrometer (JASCO, Tokyo, Japan) and Chirascan CD Spectrometer (Applied Photophysics, UK) with a temperature-controlled Peltier. Recombinant proteins at a final concentration of 2 μM were analysed using a 2-mm path-length quartz cuvette (Hellma). Spectral scans were recorded from 250 to 180 nm and averaged for least 3 scans. The SRCD data were processed and deconvoluted using the Dichroweb server [29] and the CONTINLL algorithm with the SP175 reference set [30]. To further predict the secondary structure content of PfHop, the

BeStSel server (http://bestsel.elte.hu/; [31, 32]) and the Phyre2 server (http://sbg.bio.ic.ac.uk/phyre2/; [33]) were also used. In order to investigate the heat stability of PfHop, the protein was subjected to increasing temperature (10˚C to 90˚C using 5˚C intervals for SRCD and 20 to 90˚C for CD) and full spectra recorded when the temperature was raised from 20˚C—90˚C. Similarly, spectra were recorded upon subjecting the protein to a downward shift in temperature from 90˚C to 20˚C. Melting curves were plotted by monitoring the CD signal at 192, 210 and 220 nm using the Jasco J-1500 CD spectrometer and further validated using Chirascan CD spectrometer (Applied Photophysics, UK). The CD spectral measurements were expressed as ratio of signal recorded at a particular temperature compared to the SRCD signal recorded at 10˚C. This facilitated estimation of the folded protein fraction of the protein at the respective temperature as previously described [34, 35]. The tertiary structure of the protein was probed in the presence of varying concentrations of denaturants, urea (0–8 M) and guanidine hydrochloride (0–6 M).

Fluorescence spectra were recorded following initial excitation at 295 nm and emission was determined at wavelength range of 300 nm to 400 nm using JASCO FP-6300 spectrofluorometer (JASCO, Tokyo, Japan).

## 2.4 Small-angle X-ray scattering analysis for PfHop shape determination

Synchrotron small-angle X-ray scattering (SAXS) data were collected on the EMBL Hamburg Outstation beam line P12 at PETRA III/DESY (Hamburg). PfHop (2.2 mg/ml) and buffer samples were exposed to X-rays with a wavelength of 1.240 Å for 0.045 s. Pre-processed data were further analysed with the ATSAS software package [36]. The distance distribution calculation and *ab initio* modelling were performed using GNOM [37] and the GASBOR package [38], respectively. Human Hop TPR1 (1ELW; [39]) and bakers's yeast TPR2AB domains (3QU3; [23]) were manually fitted in the *ab initio* envelope using PyMOL 2.3.2 (Schrödinger, USA). An $R_g$ value of 5.3 nm was determined visually from the linear part of the low scattering angles (0.0087–0.0715 nm$^{-1}$) using PRIMUS [40]. The $D_{max}$ for PfHop was estimated as 24 nm, also using PRIMUS.

## 3. Results

### 3.1 Oligomeric state of recombinant PfHop

Recombinant PfHop was purified using nickel affinity chromatography as previously described [18]. The protein was further purified using ion exchange and subsequently subjected to size exclusion chromatography (S1 Fig).

Hop has been reported to exist as either monomer [21, 41, 42] or dimer [43, 44], andas largely monomeric, forming weak dimers [45]. Based on size exclusion chromatography, PfHop eluted as an elongated monomer under reducing conditions. A small fraction of dimer, likely due to partial oxidation, could be seen in some batches (S1C Fig, lanes 1–5). A recent study [46] reported that PfHop exists as a dimer. However, using multi-angle light scattering coupled to size exclusion chromatography we observed that PfHop occurs as a monomer of 73 kDa (Fig 1). This is 8% larger than the calculated theoretical molecular weight 67.6 kDa. A previous independent study [20] similarly reported an apparently higher molecular weight for STI1 that was determined using gel filtration. As a control, we determined molecular weights for bovine serum albumin (73 kDa) and ovalbumin (45 kDa), which were 10% and 6%, respectively, larger than their calculated theoretical molecular weights.

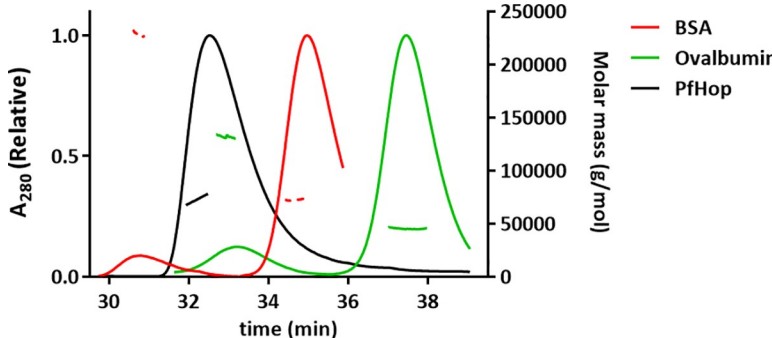

**Fig 1. Determination of the oligomeric status of PfHop.** Size exclusion chromatography of PfHop displays a single peak. The molecular weight of the peak calculated using static light scattering, shown as a black line, represents a PfHop monomer.

## 3.2 Analysis of the secondary structure of PfHop

To confirm the folding state and secondary structure composition of recombinant PfHop, synchrotron radiation circular dichroism (SRCD) spectroscopy was conducted. SRCD spectra were recorded between 175 and 280 nm at 10˚C. The PfHop spectra exhibited 2 negative minima around 222 and 208 nm and a positive peak at 194 nm (Fig 2A), characteristic of a predominantly α-helical protein [47] as previously reported [46]. Deconvolution of the spectra with Dichroweb indicated a predominantly α-helical structure comprising 77% α-helices (Table 1). This was supported by predictions from BeStSel and Phyre2 (S1 Table). The helical content estimated here for PfHop is comparable to that for *Leishmania braziliensis* Hop (LbHop) which was reported to be around 75% [42]. Notably, the PfHop helical content of 77% we observed here is much higher than that the 57% obtained for the same protein in a recent independent study [46]. While, it is not clear why such a wide discrepancy has been reported for PfHop helical content, our findings reconcile with that reported for its close homolog, LbHop [42]. Overall, the predominantly α-helical structure of PfHop is consistent with the predicted three-dimensional model of PfHop which showed that all its three TPR motifs are α-helical in nature [18]. Furthermore, based on the previously generated three-dimensional model of PfHop, residues of PfHop that are implicated in making direct contact with PfHsp70-1/PfHsp90 are positioned within the grooves of the α-helical TPR domains [18].

PfHop mediates interaction between PfHsp70-1 and PfHsp90 [18, 20], and its expression is heat-induced [18]. The roles of these two chaperones become particularly important when the parasite is subjected to physiological stress, such as during clinical malaria fever episodes [48]. It is therefore important that heat shock proteins of parasite origin exhibit resilience to heat stress conditions. PfHsp70-1 is stable at high temperatures and is most active at 48 ºC—50 ºC and is known to retain its ATPase activity at up to 80 ºC [6; 34]. However, it remains to be established whether PfHop exhibits the same resilience to heat stress. To probe this, we investigated the heat stability of recombinant PfHop in vitro. As a control, the denaturation of PfHop exposed to urea (0–8 M) was also monitored using CD (Fig 2C). Next, we monitored the folded fraction of PfHop in response to exposure to increased temperature conditions using SRCD (10˚C to 90˚C) (Fig 2A) and CD spectrometry (assay was conducted at 20˚C—90˚C; Fig 2B). PfHop appeared stable at temperatures lower than 40˚C. However, at higher temperatures the protein lost its fold, and only 50% of the protein retained its folded state at 45˚C (Fig 2C). Notably, the spectra suggest that the protein simultaneously loses both its α-helical and β-compositions in response to heat stress through two unfolding transitions ($Tm_1$ and $Tm_2$) (Fig 2B). CD spectrometry was used to monitor heat-induced denaturation of PfHop at a temperature

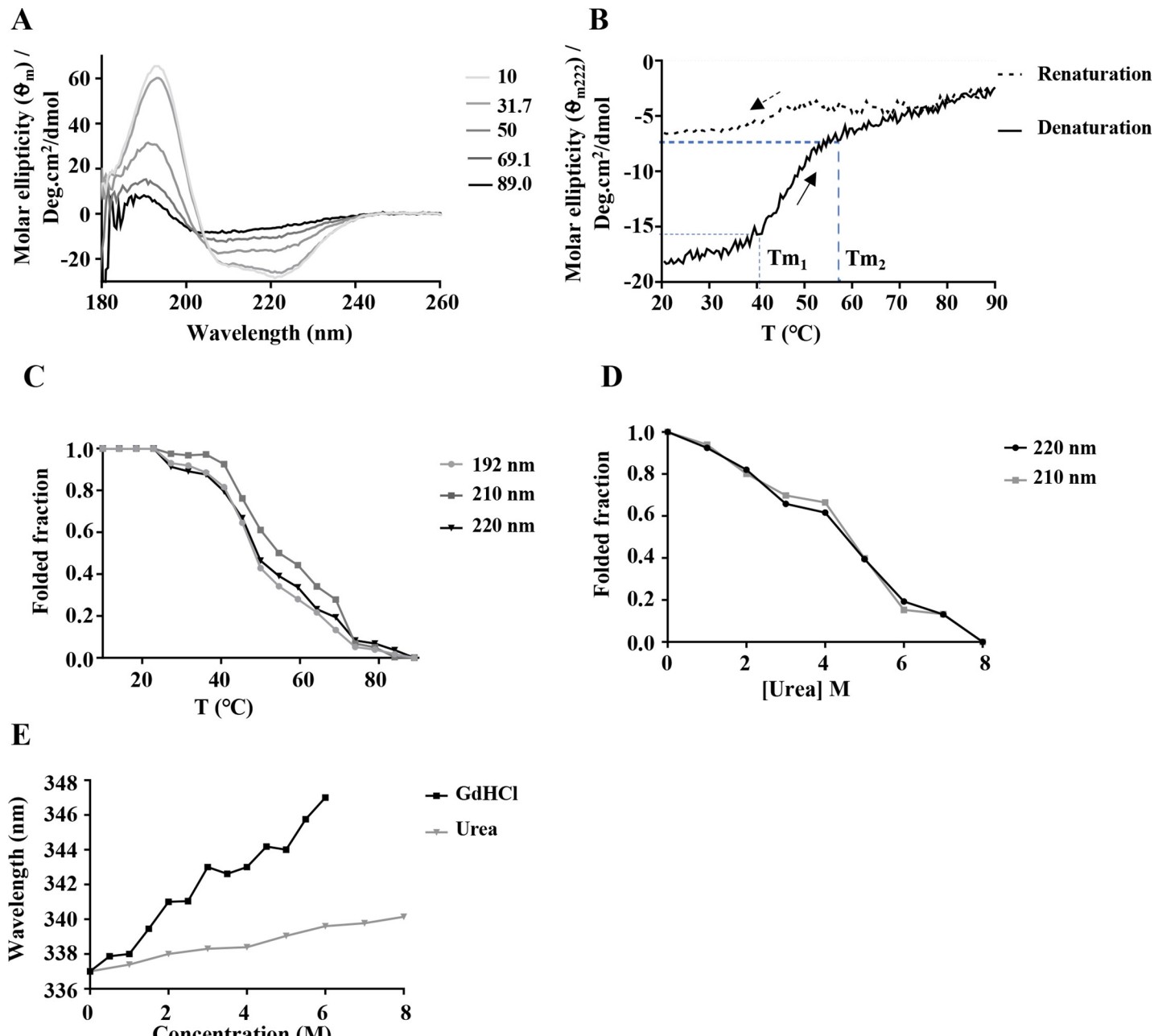

**Fig 2. Secondary structure analysis of PfHop.** (**A**) SRCD spectrum of full-length PfHop. SRCD spectral scans monitoring denaturation of PfHop upon exposure to increasing heat stress (10°C to 90°C). (**B**) Shown is the CD spectrum of PfHop monitored at 222 nm upon thermal denaturation by upscaling temperature from 20°C to 90°C. Similarly, the CD spectrum for the renaturation attempt of PfHop upon temperature downscale from 90°C to 20°C is illustrated. The thermal transitions ($Tm_1$ and $Tm_2$) are shown. (**C**) The folded fraction of PfHop as a function of temperature was monitored using CD signals at 192, 210 and 220 nm. (**D**) Urea-induced unfolding of PfHop is shown. (**E**) Represents the fluorescence emission spectra of PfHop monitored at 300–450 nm after an initial excitation at 295 nm. The recombinant PfHop protein tryptophan fluorescence emission spectra were recorded under various GdHCl and urea concentrations. Notable, is the red spectral shift obtained for of PfHop exposed to various GdHCl and urea concentrations.

range of 20°C—90°C (Fig 2B). The double phased unfolding transitions we observed here closely mirror those reported for PfHop in an independent study [46]. An attempt to refold the heat-denatured PfHop by lowering temperature from 90°C—20°C showed that the protein subjected to temperatures above 70°C could not refold. Our findings suggest that PfHop is less

**Table 1. SAXS parameters for PfHop.**

| Sample | Rg (nm) | Dmax (Å) | MW (kDa) | Expected MW (kDa) |
|--------|---------|----------|----------|-------------------|
| SLS    |         |          | 73       | 67.6              |
| SAXS   | 5.3     | 240      |          | 67.6              |

stable to heat stress than PfHsp70-1, whose ATPase activity was found to be optimal around 50˚C [34]. In addition, PfHsp70-1 exhibits chaperone activity (suppressing heat induced aggregation of protein) at 48˚C [6].

Furthermore, tryptophan fluorescence spectroscopy was conducted to monitor the tertiary structural organisation of PfHop in the presence of varying amounts of urea and guanidine hydrochloride (GdHCl). A red shift was observed with maximum peaks at 350 nm (associated with 6 M GdHCl) and 343 nm (associated with 8 M urea) (Fig 2E). PfHop was more sensitive to GdHCl, which is a stronger denaturant. This is in agreement to a previous observation for PfHsp70-1 protein [34].

## 3.3 Low-resolution structure of PfHop in solution

In order to gain further insight into the structure of PfHop, we determined its low-resolution structure in solution using SAXS (Fig 3). The X-ray scattering curve (Fig 3A), the Kratky plot (Fig 3B), and the distance distribution function (Fig 3C) together indicate that PfHop is an elongated protein with a maximum dimension of approximately 24 nm (Table 1). PfHop consists mostly of folded parts connected by flexible linkers. Thus, as expected, the TPR domains likely are arranged like pearls on a string. The distance distribution function (Fig 3C) indicates at least two stable domains with maxima at ~3 and ~5 nm within the $D_{max}$ of 240 Å. The $D_{max}$ obtained for PfHop falls within the range obtained for other Hop homologues; 180 Å for LbHop [42], 193 Å for Hop [43], 230 Å for PfHop [46] and 260 Å for STI1 [20]. Altogether, these values confirm Hop to generally assume an extended conformation across species. An *ab initio* dummy residue model calculated using GASBOR (Fig 3D) is consistent with the above data and shows an excellent fit to the experimental data (Fig 3) with a $\chi^2$ value of 1.06. The model consists of an elongated shape with some more compact regions, as observed for STI1 [20]. This model visually fits perfectly to crystal structures of human and yeast Hop TPR1 (1ELW, [39]) as well as that of the TPR2AB (3UQ3, [23]) domain (Fig 3D).

## 4. Discussion

PfHop is thought to facilitate the functional cooperation between PfHsp70-1 and PfHsp90, both prominent cytosolic molecular chaperones of *P. falciparum* [18, 19]. The Hsp70-Hop-Hsp90 pathway plays an important role in cellular development, as it facilitates folding and maturation of proteins, such as kinases and steroid hormone receptors [49]. Inhibition of both PfHsp90 and PfHsp70-1 leads to parasite death [7, 24, 26], making these molecular chaperones potential antimalarial drug targets. In the current study, we demonstrated that PfHop is unstable at temperatures above 40˚C. Furthermore, the protein is denatured at elevated temperatures through two distinct unfolding transitions (Fig 2), in agreement with a recent independent study [46]. The two unfolding transitions are consistent with a protein possessing domains of varied conformational stability. This is consistent with the multi-domain nature of PfHop as observed in the current study. In agreement with this, the N-terminus of Hop is reportedly flexible while its C-terminus is deemed to be compact [20].

We previously observed that PfHop, PfHsp70-1, and PfHsp90 occur in a complex, and that PfHop directly associates with the EEVD domains of PfHsp70-1 and PfHsp90 [18, 19]. This

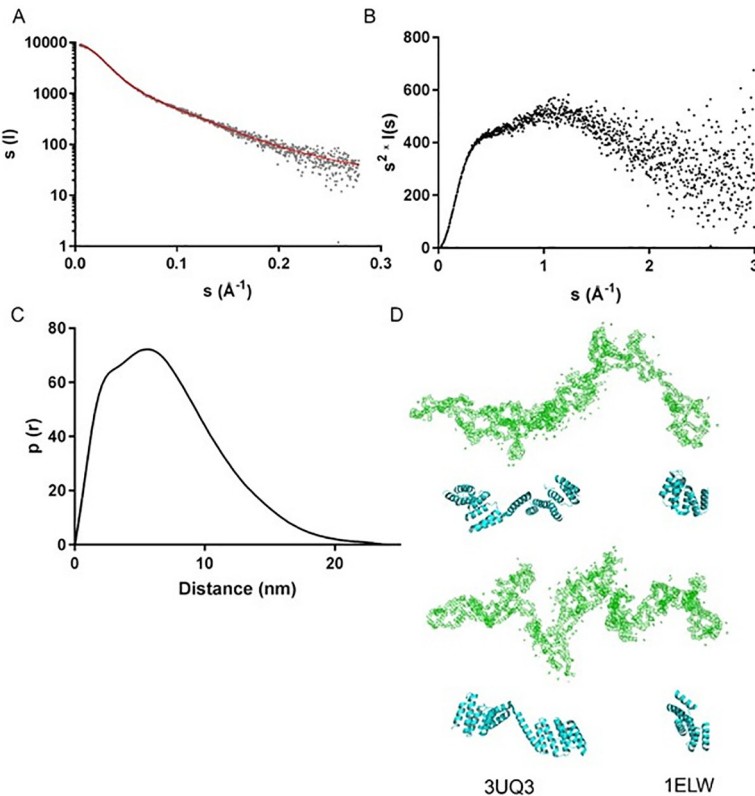

**Fig 3. SAXS analysis of PfHop.** (A) Fit of a calculated SAXS curve based on an *ab initio* model (red line) on the experimental SAXS curve (grey dots) measured for PfHop. (B) Kratky plot derived from the scattering data. (C) Distance distribution function. (D) An *Ab initio* model of PfHop (green), determined using GASBOR compared with crystal structures of human Hop TPR2AB (3UQ3) and baker's yeast TPR1 domains (1ELW) (cyan). The lower panel is related to the upper one, by a 90˚ clockwise rotation along the plane of view.

suggests that PfHop modulates functional cooperation between PfHsp70-1 and PfHsp90. The development of clinical malaria is associated with body temperature rising to 41.6˚C. At such high temperatures, the role of heat shock proteins in maintaining proteostasis becomes unquestionably important, as is evidenced by the elevated expression of key molecular chaperones, such as PfHsp70-1 and PfHsp90 [7, 50]. It is intriguing to imagine how PfHsp70-1 and PfHsp90 cooperate in the presence of limiting levels of PfHop, as occurs during sustained heat stress conditions induced by malaria fever. It is possible that under extended stress conditions, the role of Hop becomes less vital and that perhaps Hsp70 and Hsp90 may directly interact. Indeed, a study showed that despite lack of Hop in *E. coli*, Hsp70 and Hsp90 from *E. coli* are capable of direct interaction [15]. In addition, Hsp70 and Hsp90 chaperones resident in yeast cytosol and mouse ER were reported to directly bind without Hop as an adaptor [13, 14]. In a previous study, we observed complexes of PfHsp70-1 and PfHsp90 in which PfHop was present based on size exclusion chromatography of parasite lysates [18]. However, we also observed eluates representing a complex of PfHsp70-1 and PfHsp90, in which PfHop was absent [18]. A non-canonical Hop homologue from *Caenorhabditis elegans* lacks the TPR1 domain, hence it is thought to be biased towards binding to Hsp90 than Hsp70 [51]. Altogether, our findings and those of others suggest that the function of Hop may vary across species and may also depend on the prevailing cellular physiological conditions.

Findings on the oligomeric status of Hop have remained controversial as independent studies have reported it to be either monomeric [41] or dimeric [44], or largely monomeric but forming also weak dimers [45]. In the current study, we sought to establish the oligomeric status of PfHop. Based on size-exclusion chromatography and multi-angle static light scattering, we conclude that PfHop occurs as a monodisperse monomer (Fig 1). This conflicts with a recent study which reported PfHop to exist as a dimer [46]. It is important to note that we also observed a dimeric fraction which we think is on account of unspecific disulphide bridge formation that occurs under non-reducing conditions. Interestingly, our previous findings using SPR showed that PfHop self-associates with higher affinities [19], suggesting that PfHop may form oligomers. It is possible that PfHop self-association occurs transiently, hence may be detected by techniques such as SPR but not by other techniques such as SEC. This could account for the conflicting findings. On the other hand, human Hop has been reported to form dimers [44, 52, 53], but it remains to be confirmed whether these are of functional significance. In addition, Hop has also been suggested to form elongated monomers, which are difficult to resolve using gel filtration [41]. Indeed, the low-resolution solution structure determined by SAXS shows that PfHop is a highly elongated and multidomain protein. This agrees with several previous independent studies [20, 42, 43, 46]. Notably, we observed that the folded domains of PfHop are organised like beads on a string. This is consistent with the predicted concave nature of its predominantly α-helical TPR motifs [18]. TPR motifs of human Hop have been described to occur as grooves into which the C-terminal EEVD motifs of Hsp90 and Hsp70 bind in extended form [39]. Our PfHop SAXS data is consistent with this proposed model.

Altogether, our findings established that PfHop is an elongated, predominantly α-helical, monomeric, protein. Its heat stability is lower than that reported for PfHsp70-1, suggesting that its function may be compromised at high temperatures associated with clinical malaria progression.

## Supporting information

**S1 Fig. Purification of PfHop by ion exchange and size exclusion chromatography.** (**A**) SDS-PAGE analysis of the five samples collected from the main peak (lanes 1–5). Ion exchange chromatography of PfHop purification was monitored at 280 nm. Protein that bound to the column was then eluted under NaCl gradient. The five fractions obtained were pooled together for subsequent SEC analysis. (**B**) SDS-PAGE analysis of several fractions (lanes 1–11) of PfHop obtained by SEC are shown. (**C**) SDS-PAGE analysis of several fractions (lanes 1–12) of PfHop obtained by SEC are shown.
(DOCX)

**S2 Fig. Mass spectrometry sequencing data for PfHop.**
(DOCX)

**S1 Table. PfHop secondary structure content.**
(DOCX)

**S1 Raw Images.**
(PDF)

## Acknowledgments

We thank Drs. Arne Raasakka, Erik Hallin, and Juha Kallio for SRCD and SAXS data collection as well as help with data processing. We acknowledge the KIT light source for provision

of instruments at the beamline UV-CD12 of the Institute of Biological Interfaces (IBG2), and we would like to thank the Institute for Beam Physics and technology (IBPT) for the operation and storage ring, the Karlsruhe Research Accelerator (KARA). We would like to acknowledge the Biophysics, Structural Biology, and Screening (BiSS) facilities at University of Bergen for access to the static light scattering instrument. We are grateful to the office of University of Stellenbosch Central Analytical Facilities for allowing us access to use their Chirascan CD Spectrometer.

## Author Contributions

**Conceptualization:** Stanley Makumire, Inari Kursula, Addmore Shonhai.

**Data curation:** Stanley Makumire, Juha Vahokoski, Inari Kursula, Addmore Shonhai.

**Formal analysis:** Stanley Makumire.

**Funding acquisition:** Inari Kursula, Addmore Shonhai.

**Investigation:** Stanley Makumire.

**Methodology:** Stanley Makumire, Addmore Shonhai.

**Project administration:** Juha Vahokoski, Inari Kursula, Addmore Shonhai.

**Resources:** Addmore Shonhai.

**Supervision:** Tawanda Zininga, Juha Vahokoski, Inari Kursula, Addmore Shonhai.

**Validation:** Juha Vahokoski, Addmore Shonhai.

**Writing – original draft:** Stanley Makumire, Tawanda Zininga, Juha Vahokoski, Inari Kursula, Addmore Shonhai.

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
