## [Decision Letter · Decision Letter 0]

9 Jan 2020

PONE-D-19-33340

Biophysical analysis of Plasmodium falciparum Hsp70-Hsp90 organizing protein (PfHop) reveals a monomer that is characterised by folded segments connected by flexible linkers

PLOS ONE

Dear Prof. Shonhai,

Thank you for submitting your manuscript to PLOS ONE. After careful consideration, we feel that it has merit but does not fully meet PLOS ONE’s publication criteria as it currently stands. Therefore, we invite you to submit a revised version of the manuscript that addresses the points raised during the review process.

Your manuscript has now been evaluated by two expert reviewers. Their comments, which partially overlap, need to be addressed in detail. As you'll see, addressing the comments about the thermal stability experiments might also require additional data.

Moreover, previous evidence from the same lab indicating the existence of dimers should not be shrugged of as lightly as it is done in the Discussion on page 12 ("However, a dimeric fraction is observed in non-reducing conditions, indicating unspecific disulphide bridge formation, which would explain some of the previous data [12]"). This needs to be more thoroughly addressed (if necssary also experimentally).

We would appreciate receiving your revised manuscript by Feb 23 2020 11:59PM. To enhance the reproducibility of your results, we recommend that if applicable you deposit your laboratory protocols in protocols.io, where a protocol can be assigned its own identifier (DOI) such that it can be cited independently in the future. For instructions see: http://journals.plos.org/plosone/s/submission-guidelines#loc-laboratory-protocols

We look forward to receiving your revised manuscript.

Kind regards,

Didier Picard

Academic Editor

PLOS ONE

"This study has been funded by the Academy of Finland, the Norwegian Research Council, the Sigrid Jusélius Foundation, a grant (L1/402/14–1) provided to A.S. by the Deutsche Forchungsgemeinshaft (DFG) under the theme, “German–African Cooperation Projects in Infectiology”, the Department of Science and Technology/National Research Foundation (NRF) of South Africa equipment grant (UID, 75464) and NRF mobility grant (UID, 92598) awarded to A.S.; T.Z. is a recipient of the NRF Innovation Post-Doctoral fellowship UID, 111989 and African–German Network of Excellence in Science junior researcher grant."

"1. IK, Sigrid Jusélius Foundation; https://sigridjuselius.fi/en/

2. AS, L1/402/14–1; Deutsche Forchungsgemeinshaft ; https://www.validate-network.org/event/german-african-cooperation-projects-in-infectiology-dfg

3. AS; 75464; 92598; National Research Foundation/Department of Science & Innovation of South Africa; https://www.nrf.ac.za/

3. TZ; 111989; National Research Foundation of South Africa; www.nrf.ac.za"

Reviewers' comments:

Reviewer's Responses to Questions

**Comments to the Author**

1. Is the manuscript technically sound, and do the data support the conclusions?

Reviewer #1: Yes

Reviewer #2: Partly

2. Has the statistical analysis been performed appropriately and rigorously? 

Reviewer #1: Yes

Reviewer #2: Yes

3. Have the authors made all data underlying the findings in their manuscript fully available?

Reviewer #1: Yes

Reviewer #2: Yes

4. Is the manuscript presented in an intelligible fashion and written in standard English?

Reviewer #1: Yes

Reviewer #2: Yes

5. Review Comments to the Author

Reviewer #1: This manuscript from Makumire et al. examines the biophysical properties of the Hsp70/Hsp90 co-chaperone Hop from P. falciparum (PfHop). The main conclusions are:

1) PfHop is primarily monomeric

2) PfHop has folded segments connected by flexible linkers

3) PfHop is unstable at temperatures over 40C

The third conclusion is important because is suggests that PfHop may lose part of its structure/function under fever conditions associated with clinical malaria.

Most of the conclusions are well-supported although I think one control experiment is required. The authors need to establish whether the thermal denaturation is reversible or irreversible. Specifically, an experiment is needed like that shown in Fig2b where the temperature is first increased up to 80C and then decreased back down to 20C. If the denaturation is reversible then these curves should overlap. If it is not reversible then the thermal denaturation results should be interpreted cautiously. I think this is an essential control.

Minor points:

Fig 2A: CD for protein samples is usually reported as molar ellipticity (theta) with units of deg*cm^2/dmol.

Were replicate experiments performed for the thermal melt? If no, then the authors should confirm the reproducibility. Some data on reproducibility should be shown in the text or figures.

The folded fraction (y-axes) in Figures 2B&C is not well-defined because it is not clear whether 80C represent the fully denatured baseline. Also, because it is not whether the folding is cooperative or multi-state it is not clear how to convert the CD signal into a fraction folded value. I would advise plotting the y-axes as the raw CD signal.

SAXS analysis of Hop has been performed in other studies in addition to the Rohl 2015 study (for example Onuoha et al. JMB 2008). It may be helpful to briefly discuss how your SAXS results compares with previous studies.

The discussion cites literature showing Hsp90/Hsp70 directly interacting in the absence of Hop. The ER-specific Hsp70/Hsp90 (BiP/Grp94) also show a direct interaction (see Sun et al. JBC 2019).

Best Regards,

Timothy Street

Reviewer #2: In the manuscript, Makumire and colleagues use a number of biophysical techniques to analyse the Plasmodium falciparum orthologue of the Hsp70/Hsp90 co-chaperone, Hop. The authors produced the protein recombinantly in E.coli as a his-tag fusion and purified it using IMAC and ion exchange chromatography. The purified protein subsequently underwent partial biophysical characterisation using CD/SRCD, SAXS and tryptophan fluorescence. They conclude that PfHop is an elongated monomer comprised mostly of alpha helical secondary structure. In addition, they conclude that PfHop is unstable at temperatures above 40 °C, while its partner protein PfHsp70-1 is stable at much higher temperatures.

Overall the manuscript is straightforward and the data presented are consistent with previous reports, and perhaps not surprising given the sequence similarity and molecular modelling of the PfHop structure, as well as our understanding of the structural and biophysical features of Hop from other organisms. This study provides experimental data to support bioinformatics models and demonstrate that structurally PfHop is similar to Hop from other organisms. A range of appropriate biophysical methods are used in the study, and the experiments appear to have been performed appropriately. Perhaps one limitation is that all of the analyses were conducted with a single concentration of PfHop.

This study is very similar to another one from the Borges lab (doi: 10.1016/j.bbapap.2019.140282.) which has just been published. In this recent paper, the group describes the biophysical characterisation of PfHop, including a number of similar or identical methods to this study (SAXS, tryptophan fluorescence, thermal unfolding and AUC). The fact that this recent paper is so similar should not prevent publication of the current study, but it does provide a good comparison for data generated in two independent labs. The PfHop study is very recent and hence it is possible that this paper may not have been identified prior to submission which is why it is not included. Nevertheless, this study should be an essential comparison to the current manuscript.

The authors have been measured in their interpretation, and in my opinion, could perhaps improve the depth of comparison of their data to other key reports on biophysical characterisation of Hop orthologues without over-stating conclusions. The recent reports on direct interactions between Hsp70 and Hsp90 in eukaryotes needs to be given more priority and included in the introduction, not only the discussion. These studies redefine our understanding of the role that Hop plays in the Hsp70-Hsp90 complex. It is clear Hop is no longer essential for interaction in eukaryotes, although this does not mean it is not required under specific conditions. Therefore, it is not necessarily completely unexpected or surprising that PfHsp70 and PfHsp90 may be able to directly interact in the absence of PfHop.

PfHop is proposed to be instable at temperatures above 40 °C, which has implications for the PfHsp70-PfHsp90 chaperone complex during fever in malaria (where temperatures can reach this temperature). However, the conclusion of instability above 40 °C on the data presented is subjective at best. The authors should calculate the Tm which would give a more direct and comparable temperature stability. In addition, the current discussion fails to explicitly compare this with that reported for human Hop (in the region of 52 degrees C) as indicated by the study from the Regan group. This is important since the implication is that human Hop may be more stable than PfHop, which could imply biochemical differences between the two chaperone systems despite high sequence and structural similarity. In addition, the stability studies have been conducted in vitro and one needs to be cautious in directly extrapolating findings to an in vivo scenario where the protein may behave differently due to the environment. While I think the conclusion can be made, this aspect is not highlighted at all in the manuscript.

What is the relevance of the data shown in Fig 2D? How this relates to the rest of the analysis was not clearly articulated to me.

It may be better to include Fig 3D as its own figure so that it can be made larger.

The SAXS results need to be compared to the studies from the Jackson lab (Onuoha etal J Mol Biol. 2008 Jun 13; 379(4):732-44), who did SAXS on human Hop-Hsp90 complexes and is currently not referenced in the manuscript, and from the Borges lab who have done this (and other biophysical characterisation) on both LmHop (doi: 10.1016/j.abb.2016.04.008.) and PfHop.

How confident are the authors of the SEC-MAL data which appeared to consistently over-estimate the molecular weight of proteins studied? There are other studies on BSA at least which give a molecular weight at 64 kDa?

There are some minor missing articles and inconsistencies e.g. Tpr vs TPR in the manuscript that can be easily corrected with appropriate proofreading.

A minor final point, the abstract makes reference to ‘select number of cellular proteins required for cyto-protection……’. To the best of my knowledge (and I could of course be wrong), there has been limited characterisation of bona fide chaperone clients in malaria and hence, while this statement is likely true, has it been experimentally verified?

The authors are encouraged to deposit the SAXS data in the SASBDB and link the accession number to the manuscript.

6. PLOS authors have the option to publish the peer review history of their article (what does this mean?). If published, this will include your full peer review and any attached files.

Reviewer #1: Yes: Timothy Street

Reviewer #2: No

---

## [Author Response · Author response to Decision Letter 0]

27 Mar 2020

Reviewer #1: This manuscript from Makumire et al. examines the biophysical properties of the Hsp70/Hsp90 co-chaperone Hop from P. falciparum (PfHop). The main conclusions are:

1) PfHop is primarily monomeric

2) PfHop has folded segments connected by flexible linkers

3) PfHop is unstable at temperatures over 40C

The third conclusion is important because is suggests that PfHop may lose part of its structure/function under fever conditions associated with clinical malaria.

Most of the conclusions are well-supported although I think one control experiment is required. The authors need to establish whether the thermal denaturation is reversible or irreversible. Specifically, an experiment is needed like that shown in Fig2b where the temperature is first increased up to 80C and then decreased back down to 20C. If the denaturation is reversible then these curves should overlap. If it is not reversible then the thermal denaturation results should be interpreted cautiously. I think this is an essential control.

Authors Response: 

We appreciate the reviewer’s comment and we have repeated the thermal denaturation and included the data for renaturation in Fig 2b as recommended. Indeed, there is no overlap between the two curves suggesting that exposure of the PfHop to temperatures above 70°C irreversibly denatures the protein.

Minor points:

Reviewer: Fig 2A: CD for protein samples is usually reported as molar ellipticity (theta) with units of deg*cm^2/dmol.

Author’s Response: SRCD measurements were previously presented in delta epsilon and are now converted and reported as such in molar ellipticity [(θ)m]. Using the formula Δε = (θ)m * Nr / 3298 (where, Delta Epsilon = Δε; (θ)m = Molar Ellipticity (θ)m; and Nr represents the number of amino acids in the protein = 564 :

Reviewer: Were replicate experiments performed for the thermal melt? If no, then the authors should confirm the reproducibility. Some data on reproducibility should be shown in the text or figures.

Authors Response: Thermal denaturation was conducted using Synchrotron radiation circular dichroism spectroscopy. A repeat of the thermal denaturation was conducted, however using conventional CD Jasco CD and Chirascan Cd spectrometers and a similar denaturation patterns were observed. The thermal melts results from two different sets of equipment and experimental set up with 3 scans averaged showing that the thermal melt is reproducible.

Reviewer: The folded fraction (y-axes) in Figures 2B&C is not well-defined because it is not clear whether 800C represent the fully denatured baseline. Also, because it is not whether the folding is cooperative or multi-state it is not clear how to convert the CD signal into a fraction folded value. I would advise plotting the y-axes as the raw CD signal.

Authors Response: The folded fraction represents the proportion of properly folded protein at a given temperature or concentration of denaturant. The following statement from the Methods section clarifies how this assay was conducted, “The CD spectral measurements were expressed as ratio of signal recorded at a particular temperature compared to the SRCD signal recorded at 10 °C. This facilitated estimation of the folded protein fraction of the protein at the respective temperature using as previously described protocols [34; 35].” The cited reference papers have details on the functions that were used to calculate the folded fraction as follows;

The following statements were included in the methods to clarify the conversion to folded fraction, “The folded state of the proteins at any given temperature was determined as follows (Misra and Ramachandran, 2009):

([θ]t – [θ]h) / ([θ]l – [θ]h) Equation 1

Where (θ)t is the molar ellipticity at any given temperature, (θ)h at highest temperature, and (θ)l at lowest temperature, respectively.” 

“The folded state of the proteins at each urea concentration, the following equation (Eq. 1) modified to:

([θ]u - [θ]h) / ([θ]l - [θ]h) 

Where [θ]u is the molar ellipticity at any given urea concentration, [θ]h at highest urea concentration, and [θ]l at lowest urea concentration, respectively.”

Reviewer: SAXS analysis of Hop has been performed in other studies in addition to the Rohl 2015 study (for example Onuoha et al. JMB 2008). It may be helpful to briefly discuss how your SAXS results compares with previous studies.

Authors Response: The SAXS results for PfHop have been compared with those of other Hop homologues and also discussed accordingly. The following statements were included, “In order to gain further insight into the structure of PfHop, we determined its low-resolution structure in solution using SAXS (Figure 3). The X-ray scattering curve (Figure 3A), the Kratky plot (Figure 3B), and the distance distribution function (Figure 3C) together indicate that PfHop is an elongated protein with a maximum dimension of approximately 24 nm (Table 1). PfHop consists mostly of folded parts connected by flexible linkers. Thus, as expected, the TPR domains likely are arranged like pearls on a string. The distance distribution function (Figure 3C) indicates at least two stable domains with maxima at ~3 and ~5 nm within the Dmax of 240 Å . The Dmax obtained for PfHop falls within the range obtained for other Hop homologues; 180 Å for LbHop [42], 193 Å for Hop [43], 230 Å for PfHop [46] and 260 Å for STI1 [20]. Altogether, these values confirm Hop to generally assume an extended conformation across species. An ab initio dummy residue model calculated using GASBOR (Figure 3D) is consistent with the above data and shows an excellent fit to the experimental data (Figure 3) with a χ2 value of 1.06. The model consists of an elongated shape with some more compact regions, as observed for STI1 [20]. This model visually fits perfectly to crystal structures of human and yeast Hop TPR1 (1ELW, [39]) as well as that of the TPR2AB (3UQ3, [23]) domain (Figure 3D).” 

“Indeed, the low-resolution solution structure determined by SAXS shows that PfHop is a highly elongated and multidomain protein. This agrees with several previous independent studies [20; 42; 43; 46]. Notably, we observed that the folded domains of PfHop are organised like beads on a string. This is consistent with the predicted concave nature of its predominantly α-helical TPR motifs [18]. TPR motifs of human Hop have been described to occur as grooves into which the C-terminal EEVD motifs of Hsp90 and Hsp70 bind in extended form [39]. Our PfHop SAXS data is consistent with this proposed model.”

Reviewer: The discussion cites literature showing Hsp90/Hsp70 directly interacting in the absence of Hop. The ER-specific Hsp70/Hsp90 (BiP/Grp94) also show a direct interaction (see Sun et al. JBC 2019).

Authors Response: The following statement was added to the discussion citing recent literature, “Interestingly, Hsp70 and Hsp90 resident in the cytosols of E. coli and yeast as well as their homologs localized to mouse ER were shown to interact directly independent of Hop mediation [11; 12; 13; 14].” Reference 14 represents work by Sun et al., 2019 which we now included.

Reviewer #2: In the manuscript, Makumire and colleagues use a number of biophysical techniques to analyse the Plasmodium falciparum orthologue of the Hsp70/Hsp90 co-chaperone, Hop. The authors produced the protein recombinantly in E.coli as a his-tag fusion and purified it using IMAC and ion exchange chromatography. The purified protein subsequently underwent partial biophysical characterisation using CD/SRCD, SAXS and tryptophan fluorescence. They conclude that PfHop is an elongated monomer comprised mostly of alpha helical secondary structure. In addition, they conclude that PfHop is unstable at temperatures above 40 °C, while its partner protein PfHsp70-1 is stable at much higher temperatures.

Overall the manuscript is straightforward and the data presented are consistent with previous reports, and perhaps not surprising given the sequence similarity and molecular modelling of the PfHop structure, as well as our understanding of the structural and biophysical features of Hop from other organisms. This study provides experimental data to support bioinformatics models and demonstrate that structurally PfHop is similar to Hop from other organisms. A range of appropriate biophysical methods are used in the study, and the experiments appear to have been performed appropriately. Perhaps one limitation is that all of the analyses were conducted with a single concentration of PfHop.

Authors Responses: As this is a follow-up study using optimized protocols such as work from Gitau et al.,2012 [18]; Zininga et al., 2015 [19]; among other studies on the same/related proteins, the current protocol was thus guided by guidelines from such work. However, for distinct assays, we used protein concentrations that were appropriate for each assay as outlined in the methods section. Below we highlight extracts of some of the assays, highlighting protein concentrations used:

1. SRCD assays: “PfHop at a concentration of 0.5 mg/ml dialysed in buffer D (10 mM K3PO4, pH 7.0, 150 NaF) was analysed using a 98.56 µm path length round cell cuvette (Suprasil, Hellma Analytics, Germany) at a constant temperature of 10°C.”

2. SEC/SLS: 100 µg of PfHop in buffer C was injected into the column using a flow rate of 0.5 ml/min. BSA and ovalbumin were used as molecular-weight controls. The molecular weight of PfHop was determined based on the measured light scattering at three different angles and the refractive index using the ASTRA software version 6.1.5.22 (Wyatt Technology, Germany).

3. SAXS analyses: PfHop (2.2 mg/ml) and buffer samples were exposed to X-rays with a wavelength of 1.240 Å for 0.045 s. Pre-processed data were further analysed with the ATSAS software package [27].

Reviewer: This study is very similar to another one from the Borges lab (doi: 10.1016/j.bbapap.2019.140282.) which has just been published. In this recent paper, the group describes the biophysical characterisation of PfHop, including a number of similar or identical methods to this study (SAXS, tryptophan fluorescence, thermal unfolding and AUC). The fact that this recent paper is so similar should not prevent publication of the current study, but it does provide a good comparison for data generated in two independent labs. The PfHop study is very recent and hence it is possible that this paper may not have been identified prior to submission which is why it is not included. Nevertheless, this study should be an essential comparison to the current manuscript.

Authors Response: The findings from this study have now been compared to the recent PfHop paper (Silva et al., 2019 [46]) and discussions made accordingly in the various sections. Changes have been tracked.

Reviewer: The authors have been measured in their interpretation, and in my opinion, could perhaps improve the depth of comparison of their data to other key reports on biophysical characterisation of Hop orthologues without over-stating conclusions. The recent reports on direct interactions between Hsp70 and Hsp90 in eukaryotes needs to be given more priority and included in the introduction, not only the discussion. These studies redefine our understanding of the role that Hop plays in the Hsp70-Hsp90 complex. It is clear Hop is no longer essential for interaction in eukaryotes, although this does not mean it is not required under specific conditions. Therefore, it is not necessarily completely unexpected or surprising that PfHsp70 and PfHsp90 may be able to directly interact in the absence of PfHop.

Authors Responses: The data has been compared to other previous studies characterizing various Hop homologues. The direct interaction of Hsp70/Hsp90 was revisited and aspects included in the introduction. The introduction now reads as, “Stress inducible protein 1 (STI1) was first described in mouse [10], and now also known as Hsp70-Hsp90 organizing protein (Hop), acts as a module that allows Hsp70 and Hsp90 to interact stably, thereby facilitating substrate transfer from Hsp70 to Hsp90. Interestingly, there are recent findings that redefine our understanding of the role that Hop plays in the Hsp70-Hsp90 complex. Several studies have reported the direct interactions between Hsp90 and Hsp70 (Nakamoto et al., 2014; Kravats et al., 2017; 2018, Sun et al., 2019). Using Hsp90 and Hsp70 from bacteria and yeast, direct Hsp90/Hsp70 interaction was reported to occur via a region in the middle domain of Hsp90 (Nakamoto et al., 2014; Kravats et al., 2017; 2018). The ER Hsp70/Hsp90 (BiP/Grp94) were also shown to interact directly in a nucleotide dependent manner (Sun et al., 2019). Given that there is no Hop homolog in E. coli, a direct association between Hsp90 and Hsp70 in E. coli is important (Genest et al., 2016). The Hop gene is not essential in yeast (Reidy et al., 2018), Trypanosoma cruzi and Trypanosoma brucei although in the absence of Hop certain functions are affected (Schmidt et al., 2018). Hop may not be essential in some organisms but may be required under certain conditions. This might explain the need for Hsp90/Hsp70 direct interaction (Nakamoto et al., 2014). It remains to be investigated whether this direct interaction is conserved among species. Hop has been proposed as a potential drug target (Gitau et al., 2012; Zininga et al., 2015b). However, this would only apply in cases where Hop is essential for example in cancer (Röhl et al., 2014) and Leishmaniasis (Morales et al., 2010; Hombach et al., 2013).

Hop is a conserved and stress inducible protein that possesses three tetratricopeptide repeats (TPR): TPR1, TPR2A and TPR2B [10]. Both Hsp70 and Hsp90 interact with Hop via the C-terminal EEVD motif, present in the two molecular chaperones [1119; 23; 12]. Hop interacts with Hsp70 and Hsp90 via its TPR1 and TPR2A domains, respectively [1123]. While for a long time the role of the TPR2B domain of Hop has remained largely elusive, it is now thought that Hsp70 first binds to the TPR1 domain of Hop before switching to the TPR2B domain to facilitate substrate transfer to Hsp90 [20; 23].’’

Reviewer: PfHop is proposed to be instable at temperatures above 40 °C, which has implications for the PfHsp70-PfHsp90 chaperone complex during fever in malaria (where temperatures can reach this temperature). However, the conclusion of instability above 40 °C on the data presented is subjective at best. The authors should calculate the Tm which would give a more direct and comparable temperature stability. In addition, the current discussion fails to explicitly compare this with that reported for human Hop (in the region of 52 degrees C) as indicated by the study from the Regan group. This is important since the implication is that human Hop may be more stable than PfHop, which could imply biochemical differences between the two chaperone systems despite high sequence and structural similarity. In addition, the stability studies have been conducted in vitro and one needs to be cautious in directly extrapolating findings to an in vivo scenario where the protein may behave differently due to the environment. While I think the conclusion can be made, this aspect is not highlighted at all in the manuscript.

Authors Response: The PfHop Tm is the temperature at which 50 % of the protein is folded. We already have a folded fraction curve (Figure 2B shown as the temperature transitions that the protein undergoes during thermal unfolding. The calculated Tm for the protein is 50.8 °C and we prefer going with the thermal transitions of the protein unfolding.

Reviewer: What is the relevance of the data shown in Fig 2D? (now 2E) How this relates to the rest of the analysis was not clearly articulated to me.

Authors Response: Figure 2D shows the tertiary structure chemical perturbation of PfHop in the presence of denaturants (urea and guanidine chloride). This was monitored by tryptophan fluorescence to mirror the chemical perturbations observed in secondary structure analysis using CD spectra in Figure 2D. 

Reviewer: It may be better to include Fig 3D as its own figure so that it can be made larger.

The SAXS results need to be compared to the studies from the Jackson lab (Onuoha et al J Mol Biol. 2008 Jun 13; 379(4):732-44), who did SAXS on human Hop-Hsp90 complexes and is currently not referenced in the manuscript, and from the Borges lab who have done this (and other biophysical characterisation) on both LmHop (doi: 10.1016/j.abb.2016.04.008.) and PfHop.

Authors Response: The SAXS data for PfHop has been compared with other SAXS studies on various Hop homologs as suggested.

Reviewer: How confident are the authors of the SEC-MAL data which appeared to consistently over-estimate the molecular weight of proteins studied? There are other studies on BSA at least which give a molecular weight at 64 kDa?

Authors Response: We are conscious of the limitations of the SEC-MAL data and do discuss this in our MS.

Reviewer: There are some minor missing articles and inconsistencies e.g. Tpr vs TPR in the manuscript that can be easily corrected with appropriate proofreading.

Authors Response: Inconsistencies were identified and rectified as in, “Human Hop TPR1 (1ELW; 0) and bakers’s yeast TPR2AB domains (3QU3; 11) were manually fitted in the ab initio envelope using PyMOL 2.3.2 (Schrödinger, USA). Any other minor inconsistencies were corrected in the text and highlighted through the track changes option. 

Reviewer: A minor final point, the abstract makes reference to ‘select number of cellular proteins required for cyto-protection……’. To the best of my knowledge (and I could of course be wrong), there has been limited characterisation of bona fide chaperone clients in malaria and hence, while this statement is likely true, has it been experimentally verified?

Authors Response: The statement was reworded to “The cooperation of heat shock protein (Hsp) 70 and 90 is thought to facilitate folding of select group of cellular proteins that are crucial for cyto-protection and development of the parasites”.

Reviewer: The authors are encouraged to deposit the SAXS data in the SASBDB and link the accession number to the manuscript.

Authors Response: Thank you for this suggestion. We are considering depositing the data in the SASDB

---

## [Editor Report · Decision Letter 1]

6 Apr 2020

Biophysical analysis of Plasmodium falciparum Hsp70-Hsp90 organizing protein (PfHop) reveals a monomer that is characterised by folded segments connected by flexible linkers

PONE-D-19-33340R1

Dear Dr. Shonhai,

We are pleased to inform you that your manuscript has been judged scientifically suitable for publication and will be formally accepted for publication once it complies with all outstanding technical requirements.

With kind regards,

Didier Picard

Academic Editor

PLOS ONE
---

## [Editor Report · Acceptance letter]

9 Apr 2020

PONE-D-19-33340R1 

Biophysical analysis of Plasmodium falciparum Hsp70-Hsp90 organising protein (PfHop) reveals a monomer that is characterised by folded segments connected by flexible linkers 

Dear Dr. Shonhai:

I am pleased to inform you that your manuscript has been deemed suitable for publication in PLOS ONE. Congratulations! Your manuscript is now with our production department. 

With kind regards,

on behalf of

Dr. Didier Picard 

Academic Editor

PLOS ONE